# Catatonia in Systemic Lupus Erythematosus

**DOI:** 10.3390/neurosci6030090

**Published:** 2025-09-09

**Authors:** Ciro Manzo, Jordi Serra-Mestres, Marco Isetta

**Affiliations:** 1Internal and Geriatric Medicine Department, Azienda Sanitaria Locale Napoli 3 Sud, Health District No. 59, Viale dei Pini 1, Sant’Agnello, 80065 Naples, Italy; 2Psicogeriatric Outpatient Clinic, Catalunya, 17001 Girona, Spain; jordi.serra.mestres@gmail.com; 3Central and North West London NHS Trust, London NW1 3AX, UK; marco.isetta@nhs.net

**Keywords:** catatonia, systemic lupus erythematosus, neuropsychiatric systemic lupus erythematosus, narrative review, benzodiazepines, electroconvulsive therapy, anti-phosholipid antibodies, anti-ribosomal P protein antibodies, autoimmune encephalitis

## Abstract

Background: Systemic lupus erythematosus (SLE) is reported to be the most common rheumatological disorder associated with catatonia. To date, reports on catatonia manifestations in SLE patients are uncommon in published literature, which has often favored a fragmented vision. We performed a narrative review with the aim of identifying all published reports of catatonia in SLE patients to ascertain—in a comprehensive view—its clinical characteristics and to provide useful insights for daily clinical practice. Methods: Comprehensive literature searches were carried out on 10 March 2025 (subsequently repeated ahead of draft on 6 June) in all main bibliographic databases: MEDLINE and EMBASE (OVID interface); PsycINFO (ProQuest); and PubMed, to capture within-text references. All searches combined controlled (MESH, Entree, and APA Headings) and free-text elements for both areas under observation: systemic lupus erythematosus (SLE) AND catatonia, with primary focus on case reports and series. Sets of findings were reviewed separately by the authors, and the full text of selected items was sourced. Further useful references were retrieved through citation lists. Results: 39 cases of patients with SLE and catatonia were identified (35 females and 4 males), with a mean age of 22.64 years (range 11–46). Only three patients were over the age of 40; a total of 10 had catatonia at the same time of SLE onset and 5 within a month of SLE diagnosis. Antiphospholipid and anti-ribosomal P protein antibodies were rarely identified. Almost all the patients improved following treatment with lorazepam and/or electroconvulsive therapy. Only one case of malignant catatonia was reported. Finally, a large number of patients were Asian or Afro-American, at least in the reports where ethnicity was specified. Conclusions: Catatonia can occur in patients with SLE, and it may be its first clinical manifestation, especially in young patients. Its prognosis is mostly favorable.

## 1. Introduction

The syndrome of catatonia consists of signs of motor, volitional, and emotional dysfunction, often accompanied by autonomic abnormalities. Its causes are multiple, ranging from psychiatric, neurological, and general medical to drugs and substances of abuse [1,2]. Genetic abnormalities can be associated with catatonia [3,4].

Prevalence of catatonia has been reported to be 20.6% amongst those with medical and neurological etiologies [5]. On the other hand, a study in a general hospital found that 59% of patients who retrospectively met criteria for catatonia had not been diagnosed [6]. Such findings suggest that catatonia could be under-recognized and under-treated in hospital settings. The use of a rating scale such as the Bush–Francis Catatonia Rating Scale (BFCRS) can help. To date, the BFCRS is the most widely used standardized scale to assess for catatonia. In this scale, a score of 2 or more on the first 14 screening items is considered positive and should be followed by the completion of the full 23-item scale [7]. The BFCRS can also be used to monitor each clinical change. In addition, it is the only validated scale to assess the efficacy (or not) of lorazepam treatment.

It has been noted that there is an associated medical disorder contributing to the presentation in around 20% of patients with catatonia in unselected populations. It has also been suggested that some clinical features might point towards a medical cause in a patient with catatonia, namely co-morbid delirium, significant autonomic dysfunction, catatonic excitement, grasp reflex, pneumonia, history of a neurological condition, and history of seizures [8].

Very uncommon is the “malignant” variant (malignant catatonia, MC) characterized by fever, rigidity, delirium, and autonomic instability (for example, hypotension alternating with hypertension, bradycardia alternating with tachycardia, hypopnea alternating with hyperpnea, and so on). MC is a rapidly progressive condition and can be life-threatening if not diagnosed and promptly treated [9,10].

Systemic lupus erythematosus (SLE) is reported to be the most common rheumatological disorder associated with catatonia [11,12]. SLE is an autoimmune-induced multisystem disease, and neuropsychiatric manifestations are possible. The acronym NPSLE is for neuropsychiatric systemic lupus erythematosus. The pathogenesis of NPSLE includes autoantibody production more or less specific for brain structures, immune complex deposition, microangiopathy, and local release of proinflammatory cytokines. Only in about 50% of patients, however, neuropsychiatric features can be directly attributable to SLE, and this seems to be more common when antiphospholipid (aPL) or anti-ribosomal P protein (anti-rib P) antibodies are present [13,14,15]. Specifically, anti-P antibodies have high specificity for SLE, their description in other connective tissue diseases being only occasional [16].

Catatonia is not currently a classification criterion for SLE [17]. Specifically, according to the 2019 European League against Rheumatisms (EULAR) and American College of Rheumatology (ACR), only delirium, psychosis, and seizures are neuropsychiatric classification criteria, with more weight for seizures (five points) and less for delirium (two points) (Table 1).

In addition, catatonia does not even feature in the 12 central neuropsychiatric clinical syndromes identified by the 1999 ACR ad hoc committee on NPSLE nomenclature (Table 2) [18].

To date, reports on catatonia manifestations in SLE patients are uncommon in published literature, providing a fragmented vision. We performed a narrative review with the aim of identifying all published reports of catatonia in SLE patients to ascertain—in a comprehensive way—its clinical characteristics and to provide useful insights for daily clinical practice.

## 2. Materials and Methods

Comprehensive literature searches were carried out on 10 March 2025 (subsequently repeated ahead of draft on 6 June) in all main bibliographic databases: MEDLINE and EMBASE (OVID interface); PsycINFO (ProQuest); and PubMed, to capture within-text references.

All searches combined controlled (MESH, Entree, and APA Headings) and free-text elements for both areas under observation: systemic lupus erythematosus (SLE) AND catatonia, with primary focus on case reports and series.

Sets of findings were reviewed separately by the authors, and full text of selected items was sourced. Further useful references were retrieved through citation lists.

Conference abstracts, comments, and secondary articles—as well as non-English language reports with no English abstract—were excluded.

From the selected case reports, the following data were extracted: demographic details, specific signs of catatonia, medical and psychiatric history, details of the episode of SLE reported with both clinical and laboratory findings, presence of any associated psychiatric conditions, treatments for catatonia received, and the reported outcome of the case.

The catatonic signs mentioned in each case were tabulated and were checked against both the BFCRS and the *DSM-5* criteria for catatonia to ascertain whether diagnostic criteria were met.

## 3. Results

Thirty-nine cases of patients with SLE and catatonia were identified. There were 35 females and 4 males, with a mean age of 22.64 years (range 11–46). Only three patients were over the age of 40. They were all diagnosed with SLE. Twenty-one patients had a history of SLE lasting months; of these, 10 had catatonia at the same time of SLE onset; 5 within a month of SLE diagnosis; and 6 within the first 6 months following the SLE diagnosis. Sixteen cases had a diagnosis of SLE prior to the onset of catatonia, which lasted over 1 year; of these, 12 developed catatonic manifestations within the first 6 years of the SLE diagnosis. In two cases, the duration of SLE was not stated. In relation to catatonia, all 39 patients met criteria for catatonia using the BFCRS, and only 31 met the *DSM-5* criteria [19]. Out of 39, 13 presented with excitement during their illness; of these, five had a combination of hyperkinetic and hypokinetic signs. The remaining 21 patients presented with a predominance of signs in the hypokinetic variant. Only one case of MC was reported [20].

In addition, at the time of presentation, 27 patients had associated psychiatric and neuropsychiatric conditions: specifically, a depressive disorder (5), psychotic depression (2), psychosis (15), anxiety (2), mania (1), steroid psychosis (1), and delirium (1).

The most frequent catatonic signs were mutism (27 cases, 69.2%), stupor/immobility (22 cases, 56.4%), followed by rigidity (16, 41%), posturing (14, 35.9%), excitement (13–33.3%), withdrawal (13–33.3%), staring (11), stereotypy (12), waxy flexibility (12), catalepsy (10), negativism (10), echo phenomena (10), grimacing (7), autonomic dysfunction (6), verbigeration (3), combativeness (4), perseveration (3) mannerisms (3), automatic obedience (2), Mitgehen (1), and grasping (1).

All patients had received treatment for SLE as well as symptomatic treatment for catatonia with lorazepam (20 cases), lorazepam plus diazepam (2), an unnamed benzodiazepine (2), lorazepam plus midazolam (1), sodium amytal (1), ECT (5), combination of lorazepam and ECT (9), and treatment for SLE with steroids and/or immunosuppressors only (4). Of the patients receiving ECT, three received right unilateral ECT. The mean number of sessions of ECT given was 12.83 (range 4 to 22). The three cases receiving right unilateral ECT had 20 and 6 sessions, respectively. Some researchers proposed plasma exchange (PE) as a treatment approach, documenting that repeated PEs are quickly effective in SLE patients with severe and resistant catatonic manifestations, especially when associated with immunosuppressive therapy such as high-dose glucocorticoids and cyclophosphamide pulses. However, evidence for this modality of treatment comes from a very limited number of case reports, and therefore, caution should be exercised in the absence of appropriately designed studies before it can be recommended as a treatment for catatonia in SLE.

PE acts on the pathogenetic mechanisms of SLE, unlike ECT, which only acts on the catatonic manifestations [21,22]. In all cases, the documented outcome was that of recovery from or remission of catatonia. This partly confirmed what is already known for most patients with catatonia, namely that symptomatic treatment with benzodiazepines and/or ECT is effective, regardless of the underlying cause; that is, standard symptomatic treatment of catatonia is as effective in cases of SLE as it is in cases with other etiologies. However, some SLE patients benefited from immunosuppressive treatment, including high-dose glucocorticoids. The associated psychiatric conditions were treated with other medications such as antidepressants and antipsychotic drugs or with ECT.

We listed all the data in Table 3, Table 4, Table 5 and Table 6.

## 4. Discussion

Our review highlighted that catatonia is exceptionally reported in patients over 40 years of age and far from uncommon in younger patients.

Out of 39 retrieved case reports of catatonia in SLE, 15 (38.46%) patients had catatonia as an initial or early clinical manifestation: Specifically, 10 patients had catatonia as an initial manifestation, sometimes accompanying a psychosis or mood disorder; five cases had catatonia diagnosed within days to 1 month of diagnosis of SLE. Additionally, at the time of onset (or diagnosis) of catatonia, 27 patients had other associated neuropsychiatric conditions, primarily psychosis (18) or mood disorders (9). This may suggest that the risk of not recognizing catatonia could be higher when there is a concurrent psychotic or mood disorder. Psychosis and mood disorders are two of the NPSLE core features. Thus, it is possible that some patients with SLE could be diagnosed as having NPSLE without recognizing catatonia. Moreover, the risk of underdiagnosis could increase when *DSM-5* criteria are applied, as they are more restrictive, requiring the presence of at least three catatonic signs [19]. Eight patients did not meet the *DSM-5* criteria, whereas all thirty-nine patients were diagnosed by applying the BFCRS. The ability of the BFCRS compared to the *DSM-5* criteria to identify cases of catatonia has already been highlighted in a recent study [51]. However, the discrepancy in diagnostic rates between the *DSM-5* and the BFCRS may reflect differences in diagnostic sensitivity versus specificity, which may also imply misdiagnoses of catatonia when the less restrictive BFCRS is used.

It is estimated that only 30–50% of NP manifestations can be due to SLE [12,15,52]. In addition, as with the other NP syndromes, even catatonia does not have features that are SLE-specific. Therefore, a correct attribution of catatonia to SLE can be difficult in clinical practice, even more so when catatonia is one of the first clinical manifestations of SLE.

In 27 cases, catatonia occurred in the context of co-morbid psychiatric disorders. This obviously poses a complex diagnostic challenge to clinicians. This is further complicated because there are no specific guidelines for the diagnosis of SLE-related catatonia. According to some investigators, the attribution of NP manifestations to SLE should be based at least on the following three steps: (1) time of onset (the greater the interval, the lower the likelihood of causality); (2) identification and removal (if possible) of concurrent non-SLE factors; and (3) prevalence of some NP events in the general population (if a NP event is prevalent in a certain population, it would be problematic to attribute it to an uncommon disease such as SLE) [49]. In any case, (a) infections and/or metabolic disorders must be excluded; (b) magnetic resonance imaging (MRI) must be performed to rule out different and alternative diagnoses; and (c) detection of some auto antibodies (primarily, anti-riboP antibodies, anti-N-methyl-D-aspartate receptor [NMDAR] subunit 2 antibodies, and anti-aPL antibodies) should be carried out.

Screening for catatonia is paramount to identify it and then treat it as promptly as possible, but this can be a difficult endeavor, especially for non-specialist clinical staff. There have been attempts to create screening tools that, if positive, would trigger an in-depth assessment of the patient by a specialist. Recently, Luccarelli et al. proposed the Catatonia Quick Screen (CQS), which does not require a clinical examination but merely observing the patient for the presence of at least one of four common catatonic signs (excitement, mutism, staring, and posturing) [52]. The “A slime posture” tool [11] has also been proposed to train clinical staff in various settings but is more cumbersome. Consequently, the usefulness and utility of these tools in clinical practice shall be further researched and validated [53,54,55]. The application of diagnostic criteria (*DSM-5* and BFCRS) to these cases was carried out retrospectively based on the signs enumerated in each case report, so we relied on the accuracy of the semeiological descriptions provided. This could also be considered a limitation of this review.

According to our literature search, aPL and/or anti-riboP antibodies were not routinely searched for. Specifically, anti-P antibodies were assessed only in five patients with catatonic manifestations. Anti-riboP antibodies have been historically considered to have a pathogenetic role in NPSLE. However, a 2006 meta-analysis involving 1537 SLE patients highlighted that anti-P antibodies had low sensitivity (26%) and specificity (80%) for NPSLE, suggesting that their identification has limited diagnostic value [56]. Furthermore, anti-riboP antibodies were not included in the 2019 EULAR/ACR classification criteria. aPL antibodies were found only in three SLE patients with catatonic manifestations [23,41,46]. Autoimmune encephalitis is a well-established cause of catatonia, with anti-NMDAR antibodies being the most common pathogenic antibody identified [57,58]. The possibility that catatonia can be an initial manifestation in anti-NMDAR encephalitis has been reported [59]. On the contrary, according to our literature search, anti-NMDAR antibodies were never reported in SLE patients with catatonia. Finally, the possibility that catatonia may follow treatment with glucocorticoids should be taken into account in clinical practice, although it is a very rare occurrence [60,61].

From this review, we cannot comment on the natural history of catatonia without treatment, as all the cases included had received treatment (symptomatic and for SLE). The time to remission was variable, ranging from days to various months. Only one case showed an initial relapsing–remitting course until going into remission of the catatonic signs. We must point out that the uniformly positive outcomes observed in this review may relate to publication bias towards successfully treated cases.

Interestingly, a large number of patients in this review were Asian or Afro-American, at least in the reports where ethnicity was specified. This is not surprising because SLE prevalence and incidence rates are overall higher in Asian and Afro-American populations compared to Caucasian ones. The more recent epidemiological studies reaffirmed the known disparity between Black and white patients with SLE: The incidence and prevalence of SLE were more than twice as high in Black patients as in white patients [62,63,64,65]. Specifically, according to a recent comprehensive systematic analysis and modelling study, the prevalence of SLE in the general population varied from 15.9 (3.29 to 45.85) per 100,000 persons in southern Asia to 110.85 (26.74 to 314.1) per 100,000 persons in Latin America [64]. However, whilst SLE has a more severe course in Asian or Afro-American patients than Caucasian ones, there seem to be no studies on whether catatonia as one of the first clinical manifestations results in a more severe SLE course in certain populations. In addition, SLE-related catatonia has never been reported in most countries. As for today, therefore, the limited data available in the literature do not allow us to establish whether the association of ethnicity and presence of catatonia in SLE patients represents a true association or simply reflects referral patterns and publication bias. It would be very interesting to research whether ethnicity and/or socio-environmental context could affect the risk of developing catatonia in patients with SLE by influencing genetic and/or epigenetic factors.

Lastly, our review shows that catatonia has been reported very rarely in SLE patients over 40. We found only three cases: two more—case 4 and case 5—in a small case-series by Marra et al., were not included in this review because there were no specific details on the catatonic phenomenology of these two patients [21]. The reasons why catatonia is very uncommon in SLE patients over 40 are highly speculative. Generally, other cofactors being equal, a “major sensitivity” to neuroinflammation of the young patients’ brain could make the difference. Without a doubt, ad hoc studies are needed.

Our review has both strengths and limitations. The main strength is to have highlighted that catatonia manifestations can occur at the onset of SLE in young and young-adult patients. Overlooking this possibility could lead to diagnostic and therapeutic delays. In addition, since extensive studies on catatonia in SLE patients are not available, data on incidence, prevalence, and characteristics of SLE-related catatonia are fragmentary. Our review offers a comprehensive overview of the all-available data in published literature. However, given the small sample size of cases spanning decades, the statistical analysis in the review was limited to a descriptive one, and many questions may remain unanswered.

On the other hand, the lack of data is almost constant in retrospective assessments of case reports and case series, and our narrative review is no exception. Specifically, a relatively short follow-up duration in several reports and different diagnostic workups used to identify catatonia must be emphasized as a research limitation. Furthermore, this review only focused on articles in which catatonia had been identified by the relevant authors, and this only provides a small snapshot of known cases of catatonia. It is possible that there may be many cases of SLE in which catatonia was not diagnosed. Furthermore, we need to acknowledge the possibility of publication bias towards successfully treated cases. Authors of future case reports and clinical series may need to consider assessing their patients for catatonia. This review also highlights that more rigorous prospective studies are needed to ascertain the prevalence, risk factors, and best management strategies for catatonia in SLE patients.

## 5. Conclusions

Catatonia can occur in patients with SLE, and it may be one of its first clinical manifestations, especially in young patients. On the other hand, catatonia can also be due to co-morbidities in patients with SLE, and this possibility must be carefully excluded.

Consequently, greater familiarity with screening tools is highly desirable.

Finally, the choice of criteria we use in clinical practice is important: The BFCRS seems to be a more efficient, higher-performing than the *DSM-5* criteria, although this might result in some overdiagnoses of catatonia given its less restrictive nature.

## 6. Future Directions

Our review highlighted some issues to be addressed in future studies. Specifically,

(1) Why catatonia is an exceptional feature in SLE patients over 40 years of age;

(2) The relationship between catatonia and ethnicity needs to be better clarified;

(3) The lack of description of catatonia in most countries needs to be investigated;

(4) The search for antiphospholipid anti-ribosomal P protein antibodies should be part of the routine in the assessment of all SLE patients with catatonia. Their role in the pathogenesis of catatonia warrants further investigation;

(5) The complexity of the diagnosis of catatonia in SLE, especially in the presence of co-morbid primary psychiatric disorders, and the subsequent attribution of causality to one or the other.

We hope that the critical issues emerged by this narrative review will be explored in subsequent studies.

## Figures and Tables

**Table 1 neurosci-06-00090-t001:** Neuropsychiatric manifestations and their weight according to the 2019 EULAR/ACR classification criteria (mod. from [17]).

DELIRIUM 2 points
PSYCHOSIS 3 points
SEIZURE 5 points

**Table 2 neurosci-06-00090-t002:** The 1999 ACR case definitions for central NPSLE (mod. from [18]).

Acute confusional state
Mild cognitive impairment
Mood disorders
Headache
Anxiety disorder
Psychosis
Aseptic meningitis
AIDP
Seizures
Cerebrovascular disease
Myelopathy
Movement disorders

NPSLE = neuropsychiatric systemic lupus erythematosus; AIDP = acute inflammatory demyelinating polyradiculopathy.

**Table 3 neurosci-06-00090-t003:** Data on sex, age, length of illness, clinical and laboratory SLE findings.

Figure	Sex	Age	Length of Illness	Clinical and Laboratory Findings of Episode
Alao 2009 [23]	F	15	N/K	Alopecia, malar rash, low C3, anti-dsDNA+, anticardiolipin+
Ali 2014 [24]	F	20	1 yr	Polyarthralgia, oral ulcers, alopecia, ANA+, dsDNA+
Andreu 2024 [25]	F	32	None	Fever, mutism, anxiety, leukopenia, ANA+, anti-DNA+, anti-SM+, low C3
Bica 2015 [26]	F	25	13 yr	ANA+ anti-rib P+
Bica 2015 [26]	F	26	8 yr	ANA+, arthritis, malar rash, photosensitivity
Bica 2015 [26]	F	35	6 yr	ANA+, arthritis, rash, photosensitivity, hemolytic anemia
Boeke 2018 [27]	F	20	1 mo	Anemia, facial rash, fever, proteinuria, ANA+, anti-DNA+, anti-rib+, low C3C4
Boeke 2018 [27]	F	19	5 mo	Fever, rash, joint pain
Brelinski 2009 [28]	F	22	3 yr	Malar rash
Brelinski 2009 [28]	F	45	7 yr	Hemorrhagic Cerebrovascular Accident
Brelinski 2009 [28]	F	17	N/K	Fever, psychosis
Canders 2015 [29]	F	21	1 mo	Malar rash, oral ulcers, ANA+ ds-DNA+, low C4
Chaudhury 2017 [30]	F	24	N/K	ANA+, dsDNA+, cANCA+, pANCA+
Cristancho 2014 [31]	F	15	1 mo	Polyarthralgia malaise, enlarged lymph nodes, ANA +
Daradkeh 1987 [32]	F	19	2 wk	Arthritis, rash, alopecia, Raynaud
Ditmore 1992 [33]	F	27	N/K	Alopecia, rash, adenopathy, ANA+
Fam 2010 [34]	F	25	4 yr	ANA+, anti-DdsDNA+, anemia
Fricchione 1989 [35]	F	24	3 yr	Rash, discoid lesions, leukopenia, ANA+, anti-DNA+, anti-SM+, anti-RNP+
Grover 2013 [36]	F	22	2 yr	ANA+ lupus anticoagulant ab + CVA
Hussain 2015 [37]	F	30	10 yr	Not stated
Jones 2016 [38]	F	19	2 yr	Psychosis
Kim 2018 [39]	M	21	1 mo	Fever, arthralgia, malar rash, dsDNA+, low C3C4
Kronfol 1977 [40]	F	42	6 yr	Arthritis
Leon 2014 [41]	F	14	1 yr	Anticardiolipin+, antiRo+, anti-rib P+
Mac 1983 [42]	F	27	6 mo	Negative laboratory tests
Malur 1998 [43]	F	26	N/K	Lupus cerebritis
Malur 2001 [44]	F	26	N/K	Psychosis
Mamadapur 2024 [45]	M	15	None	Fever, bleeding gums, melena, pancytopenia, ANA+, anti-DNA+
Mamadapur 2024 [45]	F	14	2 yr	Catatonia, anxiety, fearfulness
Mamadapur 2024 [45]	M	15	None	Fever, altered sensorium, catatonia
Mon 2012 [46]	F	15	3 mo	ANA+, dsDNA+, SSA+, anticardiolipin+
Pai 2020 [20]	F	15	N/K	Fever, mood change, irritability, psychosis, excitement, catatonia, anti-DNA+ low C3
Perisse 2003 [22]	F	15	6 mo	ANA+, anti-dsDNA+, SSA+, antiSM+, anti-rib P+
Pustlinik 2011 [47]	F	46	N/K	ANA+, anti-dsDNA+, SSA+, thrombocytopenia, hemolytic anemia, arthritis
Sundaram 2022 [48]	F	11	2 mo	ANA+ anti-dsDNA+, fever rash low C3C4
Sundaram 2022 [48]	F	16	6 mo	ANA+, anti-dsDNA+, anemia, leucopenia
Tishler 1985 [49]	F	21	N/K	Anti-DNA+, rash, alopecia, anemia
Wang 2006 [50]	F	19	2 yr	Catatonia/no other Sxs stated
Wang 2006 [50]	F	24	2 yr	Catatonia/no other Sxs stated

N/K = not known; ys=years; mo=months; C3 = factor 3 of complement; ANA = antinuclear antibodies; SSA = Sjogren’s syndrome antibodies; C4 = factor 4 of complement; dsDNA = double strange DNA; Sm = smooth muscle; anti-rib P = anti-ribosomal P protein; ANCA = anti-neutrophil cytoplasm antibodies; CVA = cerebrovascular accident; Sxs = symptoms or signs.

**Table 4 neurosci-06-00090-t004:** Data on treatment for catatonia, outcomes, and associated psychiatric disorders.

Treatment for Catatonia	Outcome	Associated Psychiatric Disorders
Lorazepam	Remission	Depression, psychosis, mania
Clonazepam, fluoxetine	Recovering	Depression
Lorazepam, olanzapine, steroids, hydroxychloroquine	Remission	None
Bilateral ECT x 4	Remission	Depression, suicidal
Bilateral ECT x 12	Remission	Psychosis, confusion
Bilateral ECT x 10	Remission	Psychosis
Lorazepam, bilateral ECT x 7	Remission	Psychosis (steroid-induced)
Lorazepam	Remission	Psychosis
Lorazepam	Remission	Psychosis
Lorazepam, citalopram	Remission	Depression, psychosis, mania
Lorazepam, sulpiride	Remission	Psychosis
Lorazepam, hydroxychloroquine, cyclophosphamide	Remission	Psychosis
Methylprednisolone	Remission	None
Lorazepam, right unilateral ECT x 20	Remission	None
Prednisolone, azathioprine	Remission	Anxiety
Bilateral ECT x 11, lithium	Remission	Mania with psychosis
Lorazepam, haloperidol, olanzapine, quetiapine, bilateral ECT x 6	Remission	Delirium
Lorazepam, bilateral ECT x 17, steroids, carbamazepine	Remission	Psychosis
Lorazepam	Remission	None
Lorazepam, haloperidol, amisulpiride, bilateral ECT x 6	Remission	Psychosis
Lorazepam, midazolam, bilateral ECT x 17	Remission	Psychosis
Lorazepam	Remission	None
Sodium amytal x 1, Prednisone	Remission	Anxiety
Lorazepam, bilateral ECT x 11	Remission	Psychosis
bilateral ECT x 5	Remission	Depression
Lorazepam, bilateral ECT x 14	Remission	None
Lorazepam, right unilateral ECT x 22	Recovering	Psychosis
Lorazepam, phenytoin, cyclophosphamide	Remission–Relapse–Remission	None
Lorazepam, methylprednisolone, cyclophosphamide	Remission	None
Lorazepam, methylprednisolone, cyclophosphamide	Remission	None
Lorazepam, bilateral ECT x 20	Remission	Psychosis
Lorazepam, right unilateral ECT x 6, methylprednisolone, beta-blockers, plasma exchange	Remission	Psychosis
Benzodiazepines, antidepressant, atypical antipsychotic, methylprednisolone, cycloclophosphamide plasma exchange	Remission	Psychotic depression
Benzodiazepines, methylphenidate, escitalopram, hydoxychloroquine, low-dose prednisolone	Remission	Depression
Lorazepam, bilateral ECT x 13	Remission	None
Lorazepam, aripiprazole	Remission	Psychosis
Prednisone	Remission	Psychosis
Lorazepam, diazepam	Remission	None
Lorazepam, diazepam	Remission	None

ECT = electroconvulsive therapy.

**Table 5 neurosci-06-00090-t005:** Catatonic manifestations in SLE patients, from stupor/immobility (S/I) to verbigeration (V).

First Author and Year of Publication	Sex	Age	S/I	C	W/F	M	N	P	M	S	A/E	G	Elal	Eprax	St	V
Alao 2009 [23]	F	15	X	X		X										
Ali 2014 [24]	F	20	X	X		X									X	
Andreu 2024 [25]	F	32	X			X				X					X	
Bica 2015 [26]	F	25								X	X	X				
Bica 2015 [26]	F	26	X			X						X				
Bica 2015 [26]	F	35				X				X	X					
Boeke 2018 [27]	F	20			X			X						X	X	
Boeke 2018 [27]	F	19				X										X
Brelinski 2009 [28]	F	22					X	X			X					
Brelinski 2009 [28]	F	45	X					X	X	X						
Brelinski 2009 [28]	F	17	X							X					X	
Canders 2015 [29]	F	21	X	X		X									X	
Chaudhury 2017 [30]	F	24	X		X	X		X		X		X			X	
Cristancho 2014 [31]	F	15			X			X			X	X				
Daradkeh 1987 [32]	F	19	X		X	X					X					
Ditmore 1992 [33]	F	27				X	X	X			X					
Fam 2010 [34]	F	25								X	X		X	X		
Fricchione 1989 [35]	F	24	X			X					X	X				
Grover 2013 [36]	F	22	X			X	X						X			
Hussain 2015 [37]	F	30				X	X								X	
Jones 2016 [38]	F	19						X			X					X
Kim 2018 [39]	M	21	X			X		X		X	X				X	
Kronfol 1977 [40]	F	42	X		X	X			X	X		X			X	
Leon 2014 [41]	F	14		X							X		X	X		
Mac 1983 [42]	F	27	X													
Malur 1998 [43]	F	26				X	X									
Malur 2001 [44]	F	26	X			X							X			
Mamadapur 2024 [45]	M	15	X		X	X		X				X				
Mamadapur 2024 [45]	F	14	X		X	X	X	X	X	X			X	X		
Mamadapur 2024 [45]	M	15	X			X	X	X		X						
Mon 2012 [46]	F	15		X	X	X		X								
Pai 2020 [20]	F	15	X	X	X	X					X		X		X	
Perisse 2003 [22]	F	15	X	X	X			X								X
Pustlinik 2011 [47]	F	46		X		X										
Sundaram 2022 [48]	F	11	X	X	X	X				X	X					
Sundaram 2022 [48]	F	16	X		X	X	X			X					X	
Tishler 1985 [49]	F	21				X		X							X	
Wang 2006 [50]	F	19	X			X	X									
Wang 2006 [50]	F	24	X	X		X	X									

F= females; M= males. S/I = stupor/immobility; C = catalepsy; W/F = waxy/flexibility; M = mutism; N = negativism; P = posturing; M = mannerisms; S = stereotypy; A/E = agitation/excitement; G = grimacing; Elal = echolalia; Eprax = echopraxia; S = staring; V = verbigeration. X= present.

**Table 6 neurosci-06-00090-t006:** Catatonic manifestations in SLE patients, from rigidity to combativeness.

First Author and Year of Publication	Sex	Age	Rigidity	Withdrawal	Impulsivity	Mitgehen	Ambitendency	Perseveration	Autonomic Dys	Automatic Obedience	Gegenhalten	Grasping	Combativeness
Alao 2009 [23]	F	15	X							X			
Ali 2014 [24]	F	20	X	X									
Andreu 2024 [25]	F	32											
Bica 2015 [26]	F	25											
Bica 2015 [26]	F	26											
Bica 2015 [26]	F	35											
Boeke 2018 [27]	F	20						X					X
Boeke 2018 [27]	F	19											
Brelinski 2009 [28]	F	22											
Brelinski 2009 [28]	F	45										X	
Brelinski 2009 [28]	F	17	X										
Canders 2015 [29]	F	21	X										
Chaudhury 2017 [30]	F	24		X				X					
Cristancho 2014 [31]	F	15		X					X				
Daradkeh 1987 [32]	F	19											
Ditmore 1992 [33]	F	27		X									X
Fam 2010 [34]	F	25											
Fricchione 1989 [35]	F	24	X						X				
Grover 2013 [36]	F	22	X	X				X					
Hussain 2015 [37]	F	30	X	X									
Jones 2016 [38]	F	19				X			X		X		
Kim 2018 [39]	M	21		X									
Kronfol 1977 [40]	F	42	X							X	X		
Leon 2014 [41]	F	14	X						X				
Mac 1983 [42]	F	27		X									
Malur 1998 [43]	F	26											
Malur 2001 [44]	F	26	X										
Mamadapur 2024 [45]	M	15		X									
Mamadapur 2024 [45]	F	14											
Mamadapur 2024 [45]	M	15	X	X									
Mon 2012 [46]	F	15	X	X									
Pai 2020 [20]	F	15	X						X				X
Perisse 2003 [22]	F	15	X										
Pustlinik 2011 [47]	F	46	X										
Sundaram 2022 [48]	F	11											
Sundaram 2022 [48]	F	16	X						X				
Tishler 1985 [49]	F	21		X									X
Wang 2006 [50]	F	19		X									
Wang 2006 [50]	F	24											

F= females; M= males; X= present.

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
