# Peer review of "Catatonia in Systemic Lupus Erythematosus"

_neurosci, 2025, doi:10.3390/neurosci6030090_

Round 1
Reviewer 1 Report
Comments and Suggestions for Authors
Dear authors,
I have now completed the review of the manuscript titled Catatonia in Systemic Lupus Erythematosus
The manuscript is interesting and, in general, fairly well-written. However, I still have some suggestions to further improve the quality of the manuscript.
I would like to suggest that the authors address these limitations in the article, either by discussing them in the limitations section or, where feasible, by making the appropriate revisions:
1. The statistical analysis is extremely limited for a dataset of 39 cases spanning several decades. The authors present basic descriptive statistics but fail to conduct any meaningful comparative analyses. They note demographic patterns, such as the predominance of young patients and certain ethnic groups, but provide no statistical testing to determine if these observations are significant or merely reflect the underlying demographics of SLE populations. The comparison between Bush-Francis Catatonia Rating Scale (BFCRS) and DSM-5 criteria reveals that eight patients did not meet DSM-5 criteria while all met BFCRS criteria. However, the authors conclude that BFCRS is "higher performing" without considering that this discrepancy might reflect differences in diagnostic sensitivity versus specificity, or that some cases may have been misdiagnosed using the less restrictive BFCRS criteria.
2. The authors claim that catatonia can be the "first clinical manifestation" of SLE in 38.46% of cases, but this interpretation is problematic. Many of these patients had concurrent psychiatric symptoms or other SLE manifestations, making it unclear whether catatonia was truly the initial presentation or simply the most prominent symptom that led to medical attention. The discussion of treatment outcomes lacks critical analysis. While the authors note that all patients showed improvement with standard treatments (benzodiazepines and electroconvulsive therapy), they fail to address several important questions: What was the natural history without treatment? How long did recovery take? Were there any treatment failures or relapses? The uniformly positive outcomes described seem implausibly optimistic and may reflect publication bias toward successful cases.
3. The authors acknowledge that antiphospholipid and anti-ribosomal P protein antibodies were "rarely identified" but do not adequately explore this finding. If these antibodies are thought to be pathogenic in neuropsychiatric SLE, their absence in most catatonia cases raises important questions about disease mechanisms that the authors do not address. The geographic and ethnic distribution observations are mentioned but not rigorously analyzed. The authors note that many patients were Asian or African American but fail to control for the baseline prevalence of SLE in these populations or consider whether this represents a true association or simply reflects referral patterns and publication bias.
4. The authors' conclusions extend beyond what their data can reasonably support. They recommend increased screening for catatonia in SLE patients and suggest diagnostic approaches without providing evidence for the effectiveness of such screening programs. The clinical recommendations appear premature given the limited and biased nature of the available evidence. The study would have been strengthened by acknowledging these limitations more explicitly and tempering their conclusions accordingly. While the research provides a useful compilation of existing case reports, it falls short of establishing definitive clinical guidelines or pathophysiological insights about catatonia in SLE patients. This review highlights the need for more rigorous prospective studies to determine the true prevalence, risk factors, and optimal management strategies for catatonia in SLE patients, rather than relying on the inherently limited case report literature.
5. The relationship between catatonia and other neuropsychiatric manifestations receives superficial treatment. The authors note that 27 patients had concurrent psychiatric conditions but do not adequately address the complex differential diagnosis challenges this presents, particularly regarding attribution of symptoms to SLE versus primary psychiatric disorders. The discussion of plasma exchange as a treatment modality lacks critical evaluation of the evidence quality and appears to overstate the therapeutic implications based on limited case reports.
Thank you for your valuable contributions to our field of research. I look forward to receiving the revised manuscript.
Author Response
Dear Reviewer,
thank You for sending us these helpful and thoughtful comments and suggestions.
We hope they will answer the questions and issues raised.
Please see the attachment. Thank You very much.

Reviewer 2 Report
Comments and Suggestions for Authors
Line 40: "Prevalence of catatonia has been reported to be 20.6% amongst those with medical and neurological aetiologies [5]." This doesn't really make sense. That would mean 1 in 5 patients with neurologic conditions have catatonia? I'd make sure you are interpreting this study correctly.
Spelling error: Table II: Movement disorders
Methods section should better describe what information/data was abstracted from each case report/series, and how authors plan to report this data.
Lines 100-107: shouldn't start sentences with numerical value. Type out the number (eg. 21 should be twenty-one.
Lines 138-141. "This point deserves to be highlighted: as for most patients with catatonia from other causes, in patients with SLE treatment with benzodiazepines and/or ECT was effective. In other words, failure to respond to traditional therapies was not an element of suspicion for SLE." I'm very confused what the point is you are trying to get across. Are you trying to say "patients with catatonia from SLE responded similarly to benzodiazepines/ECT as patients with catatonia from other etiologies?" Consider rewording this to be more concise.
Table 3. Many inconsistencies of abbreviations (ds-DNA, dsDNA, anti-DNA), misspellings- Raymaud (Daradekh 1987), anti Rib+ . Some lines also describe the NP features, while others only include extra-NP features of lupus- for example you write catatonia in some lines (aren't they all catatonic) or will write mood change, psychosis, etc, but others you just put "arthritis", etc.
First paragraph of discussion lines 154-166 especially, excessive use of transitional phrases that make it sound wordy and not professional work. Significantly... Indeed... As it is known... In short... Indeed... Also used in other areas of text unnecessarily. Also be careful using a leading phrase "Significantly" when no statistical analysis was performed.
Also this paragraph using DSM-5 criteria and DSM-V criteria, inconsistent.
Author Response
Please see the attachment,

Reviewer 3 Report
Comments and Suggestions for Authors
The issue is that 31 cases met DSM-5 criteria
39 met Bush Francis, thus DSM-5 and probably ICD-11 miss cases of catatonia.
The correct reference is.
ErdoÄŸan IM, Aytulun A, AvanoÄŸlu KB, et al. Evaluation of Catatonia in the Psychiatry and Neurology Inpatient Units using Different Assessment Scales. Turkish Journal of Psychiatry, 2024 35(3), 198
The authors have carefully identified causes of catatonia. Now they must recognize that DSM-5- Criteria misses 10-20% of cases of catatonia.
This is an article by very precise researchers, yet they rely on DSM-5 criteria, which is probably good for screening but not as precise at BFCRS
Round 2
Reviewer 1 Report
Comments and Suggestions for Authors
All comments addressed.
Reviewer 2 Report
Comments and Suggestions for Authors
Overall better